# One-Step Graph-Structured Neural Flows for Irregular Multivariate Time Series Classification

Mengzhou Gao [1]   Kaiwei Wang [1]   Pengfei Jiao [1]

## Abstract

Neural Flows efficiently model irregular multivariate time series by directly learning ODE solution trajectories with neural networks, bypassing step-by-step numerical solvers. Despite their efficiency, many existing approaches treat variables independently, leaving inter-variable interactions underexplored. Moreover, their one-step mapping makes interaction modeling inherently challenging, as it removes the iterative refinement of interactions during learning. To address this challenge, we propose one-step Graph-Structured Neural Flows (GSNF), which introduce two auxiliary-trajectory self-supervision strategies to strengthen interaction learning: (i) interaction-aware trajectory generation via re-initialization, which induces trajectory divergence to expose graph-induced interactions, with a theoretically derived lower bound on divergence; and (ii) reverse-time trajectory generation, which enforces forward–backward consistency to regularize graph learning, enabled by flow invertibility. Experiments on five real-world datasets show that GSNF achieves state-of-the-art classification performance with highly competitive training time and memory usage. The code is available at https://github.com/mzgaooo/GSNF.

## 1. Introduction

Irregular multivariate time series arise from event-driven and heterogeneous data acquisition processes, leading to non-uniform sampling intervals and missing observations, which challenge conventional time series analysis methods. (Xiao et al., 2024) Despite such irregularity, the underlying system dynamics are typically assumed to evolve continuously over time, making continuous-time models a natural framework

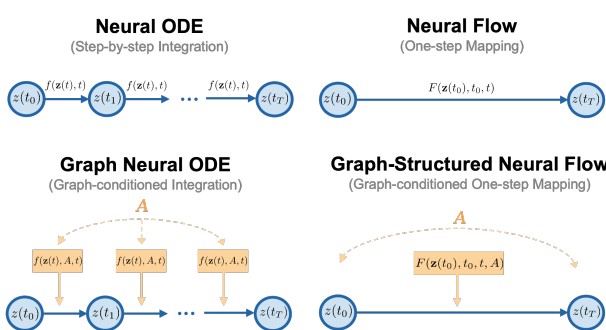

*Figure 1.* Comparison of continuous-time models. Neural ODEs evolve latent states via step-by-step numerical integration, whereas Neural Flows perform one-step mappings. Graph Neural ODEs incorporate interaction modeling by conditioning dynamics on an interaction graph, while Graph-Structured Neural Flows integrate graph-conditioned interactions into one-step mappings.

for analyzing such data. (Chen et al., 2023)

Existing continuous-time models for irregular multivariate time series can be broadly categorized into implicit and explicit formulations. (Oh et al., 2025b) Implicit methods, including Neural ODEs (Chen et al., 2018), Neural CDEs (Kidger et al., 2020), and Neural SDEs (Kidger et al., 2021), define latent dynamics through differential equations and rely on numerical solvers to evolve system states. While this formulation offers flexibility in handling irregular observations, the reliance on step-by-step numerical integration introduces non-negligible computational and memory overhead (Oh et al., 2025a). In contrast, explicit approaches such as Neural Flows (Biloš et al., 2021) learn a one-step mapping that directly models ODE solution trajectories with neural networks, thereby avoiding iterative integration and enabling more efficient computation.

While continuous-time models effectively handle irregular observations, many existing approaches treat variables independently, leaving inter-variable interactions underexplored in multivariate settings (Mercatali et al., 2024). Within the Neural ODE framework, this gap has been partially addressed by incorporating graph structures into continuous-time dynamics, enabling joint modeling of multiple interacting time series. For example, Graph Neural ODEs (Poli et al., 2019) parameterize ODE vector fields with graph

[1]School of Cyberspace, Hangzhou Dianzi University. Correspondence to: Pengfei Jiao <pjiao@hdu.edu.cn>.

*Proceedings of the 43rd International Conference on Machine Learning*, Seoul, South Korea. PMLR 306, 2026. Copyright 2026 by the author(s).

neural networks, introducing relational inductive biases into continuous-time dynamics.

However, incorporating graph-structured interactions into Neural Flows in a way that directly governs trajectory evolution remains largely unexplored. By collapsing continuous-time dynamics into a one-step mapping, Neural Flows remove numerical solvers and their intermediate latent states, causing interaction effects to be applied only once rather than iteratively refined along the trajectory. This limits incremental correction and reduces the ability to progressively refine interactions during trajectory evolution. This challenge is further amplified when graph structures must be learned under missing observations, where interaction learning and trajectory prediction are jointly resolved within a one-step formulation.

To address this challenge, we propose **G**raph-**S**tructured **N**eural **F**lows (GSNF), a framework that strengthens interaction learning in Neural Flows while retaining the efficiency of one-step mappings. GSNF introduces auxiliary trajectory-level self-supervision to compensate for the absence of iterative interaction refinement. These auxiliary signals encourage interaction-induced trajectory divergence and forward–backward temporal consistency, and can be incorporated into a VAE-based framework with parallel computation, enabling efficient end-to-end optimization of latent states and graph structures. Our main contributions are summarized as follows:

- We propose GSNF, a Graph-Structured Neural Flow that intrinsically embeds inter-variable interactions within a one-step formulation.

- We design auxiliary trajectory-level self-supervision for GSNF, including interaction-aware trajectory generation (ITG) and reverse-time trajectory generation (RTG), with a theoretical lower bound on the divergence induced by ITG.

- Experiments on five real-world datasets show that GSNF achieves state-of-the-art performance and is the most efficient among top-performing methods.

## 2. Related work

### 2.1. Continuous-Time Models for Irregular Time Series

Modeling irregularly sampled time series has long been a challenging problem, primarily due to non-uniform observation intervals and asynchronous measurements. Neural Ordinary Differential Equations (Neural ODEs) (Chen et al., 2018) first introduced a continuous-depth framework for modeling such data. Latent ODE and GRU-ODE-Bayes (De Brouwer et al., 2019) extended Neural ODEs by

introducing discontinuities at observation points, thereby improving predictive accuracy on irregularly sampled sequences. However, these methods rely on numerical solvers, which limits computational efficiency.

Neural Flows (Biloš et al., 2021) addressed this limitation by directly parameterizing solution trajectories with neural networks, removing the dependency on ODE solvers and enabling one-step trajectory computation. Building on this idea, IVP-VAE (Xiao et al., 2024) employed a single invertible initial value problem to achieve parallel continuous-time modeling, offering better scalability and efficiency compared to RNN-based ODE methods. DualDynamics (Oh et al., 2025b) further combined Neural Differential Equation (NDE)-based solvers with Neural Flows to enhance the expressive power of continuous models. Despite these advances, existing methods lack explicit modeling of inter-variable dependencies that drive trajectory evolution.

### 2.2. Graph Learning in Continuous-Time Models

In multivariate time series, the complex interdependencies among variables have motivated the integration of graph structures into continuous-time modeling frameworks. LG-ODE (Huang et al., 2020) was one of the earliest attempts to incorporate graph neural networks (GNNs) into Neural ODEs, enabling neighborhood information aggregation to improve variable modeling. Coupled Graph ODE (Huang et al., 2021) extended this idea to edge-level dynamics, while GG-ODE (Huang et al., 2023) further generalized it to multi-environment settings. However, these methods rely on multi-step numerical integration to propagate information across the graph, resulting in high computational costs and incompatibility with Neural Flow models that emphasize efficient one-step computation.

Alternative approaches have attempted to infer graph structures directly from data. For instance, Raindrop (Zhang et al., 2022) learns a dependency graph by averaging time-varying attention weights across timestamps and pruning weak connections. Similarly, GNeuralFlow (Mercatali et al., 2024) employs a directed acyclic graph (DAG) to represent conditional dependencies among variables and jointly learns this structure alongside Neural Flows, offering flexibility for modeling unknown interaction graphs. Notably, the graph in GNeuralFlow is only used to parameterize the initial condition, and does not directly participate in the subsequent flow-based trajectory generation.

## 3. Preliminaries

### 3.1. Neural Flows

Neural Flows (Biloš et al., 2021) directly model the solution trajectory of an ODE with a neural network,

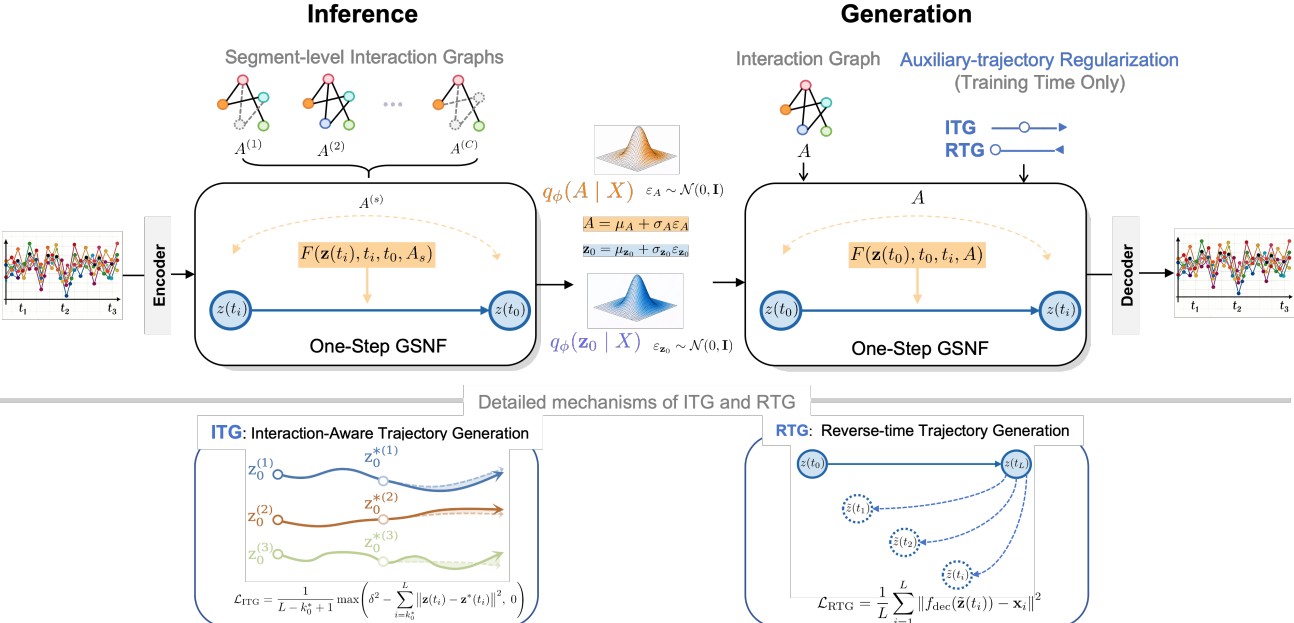

*Figure 2.* Overall framework. **Top**: Inference and generation with one-step GSNF. *During inference*, segment-level interaction graphs $\{A^{(s)}\}_{s=1}^{C}$ are aggregated into a posterior $q_\phi(A \mid X)$, and latent states are propagated backward through GSNF, conditioned on their corresponding segment-level graphs to infer $q_\phi(\mathbf{z}_0 \mid X)$. *During generation*, samples of $A$ and $\mathbf{z}_0$ are used by GSNF to compute latent states forward at arbitrary time points via one-step mappings, with auxiliary trajectory-level regularization applied during training. **Bottom**: Detailed mechanisms of ITG and RTG. *ITG* induces divergence between the original trajectory $\mathbf{z}(t)$ and the re-initialized trajectory $\mathbf{z}^*(t)$ via intermediate-time re-initialization, regularized by the trajectory-level loss $\mathcal{L}_{\text{ITG}}$. *RTG* enforces forward–backward consistency between the forward trajectory $\mathbf{z}(t)$ and the reverse-time trajectory $\tilde{\mathbf{z}}(t)$ through the reconstruction loss $\mathcal{L}_{\text{RTG}}$.

$$\mathbf{z}(t) = F(\mathbf{z}_0, t_0, t), \quad \mathbf{z}(t_0) = \mathbf{z}_0, \quad (1)$$

instead of parameterizing the derivative as in Neural ODEs (Chen et al., 2018). This eliminates the need for numerical ODE solvers, providing a faster alternative to the ODE. To guarantee correspondence with an underlying ODE, $F$ must (i) satisfy the initial condition $F(\mathbf{z}_0, t_0, t_0) = \mathbf{z}_0$, anchoring the trajectory to the specified starting point; and (ii) be invertible in $\mathbf{z}$ for all $t$, ensuring uniqueness of trajectories without self-intersections.

### 3.2. Problem Formulation

Consider a labeled multivariate time series $(X, y)$, where $y$ denotes the categorical label and $X = \{(\mathbf{x}_k, t_k, \mathbf{m}_k)\}_{k=1}^{L}$ is a sequence of $L$ irregularly sampled observations. Each observation $\mathbf{x}_k \in \mathbb{R}^{D_x}$ is recorded at non-uniform time points $t_k$ and is associated with a binary mask $\mathbf{m}_k \in \{0,1\}^{D_x}$, where $m_{k,i} = 1$ denotes that the $i$-th variable is observed at time $t_k$ and $m_{k,i} = 0$ indicates a missing value. The dataset is given by $\mathcal{X} = \{(X_n, y_n)\}_{n=1}^{N}$, consisting of $N$ such sequences.

We adopt a latent continuous-time modeling framework in which the observed sequence $X$ is assumed to arise from interacting latent dynamics $\mathbf{z}(t) \in \mathbb{R}^{D_z}$ governed by an interaction structure $A$. Our goal is to jointly infer the latent initial state $\mathbf{z}_0$ and the interaction structure $A$, and to integrate them into the Neural Flow for downstream classification. Specifically, we aim to learn a classifier $f : X \to \mathcal{Y}$ such that the predicted label

$$\hat{y} = f(\mathbf{z}_0, A, F(\mathbf{z}_0, A, t))$$

matches the ground-truth label $y$ as accurately as possible.

## 4. Graph-Structured Neural Flows

To model continuous-time dynamics with explicit interaction structure, we propose *Graph-Structured Neural Flows* (GSNF) as the core dynamical module of our framework. Unlike standard neural flows (Biloš et al., 2021) that evolve latent states independently, GSNF conditions the flow on an interaction graph $A$, allowing interactions to influence the system evolution throughout the flow.

Formally, GSNF defines a Neural Flow that maps an initial latent state at time $t_0$ to a future state at time $t$,

$$\mathbf{z}(t) = F(\mathbf{z}(t_0), t_0, t, A). \quad (2)$$

In this work, we implement GSNF following the ResNet flow architecture (Biloš et al., 2021). Unlike GNeu-ralFlow (Mercatali et al., 2024), which incorporates the

graph only to initialize the latent state, GSNF embeds graph-structured interactions directly into the flow dynamics, allowing interactions to modulate state evolution throughout the trajectory. Specifically, the flow mapping is defined as

$$F(\mathbf{z}(t_0), t_0, t, A) = \mathbf{z}(t_0) + \varphi(t - t_0)\, g(\mathbf{z}(t_0), t_0, t, A),$$
$$g(\mathbf{z}(t_0), t_0, t, A) = \mathrm{MLP}(\mathbf{z}(t_0) \| t \| t_0)$$
$$\odot \; \mathrm{GCN}(A, \mathbf{z}(t_0) \| t \| t_0), \tag{3}$$

where $\varphi(\cdot)$ satisfies $\varphi(0) = 0$, ensuring the initial condition $F(\mathbf{z}(t_0), t_0, t_0, A) = \mathbf{z}(t_0)$ and is instantiated with a Time-Fourier encoding. The operator $\odot$ denotes the element-wise product, and $\|$ denotes row-wise concatenation.

A key property of GSNF is its invertibility, which enables well-defined forward and reverse mappings. We provide sufficient conditions in the following theorem.

**Theorem 4.1** (Invertibility of Graph-Structured Neural Flows). *The GSNF defined in* (3) *is invertible if* $\varphi(\cdot) \in [0, 1)$ *and* $g(\cdot, t_0, t, A)$ *is a contractive mapping, which can be ensured by applying spectral normalization to all linear layers in both the MLP and the GCN. The proof is provided in Appendix A.*

## 5. Overall Framework

Building upon the previously introduced GSNF, our framework (Fig. 2) adopts a latent-variable (VAE) structure to jointly infer latent initial states $\mathbf{z}_0$ and the interaction graph $A$ from irregularly sampled multivariate time series. The inferred states $\mathbf{z}_0$ and graph $A$ are then used to evolve latent trajectories in *one-step via GSNF*, capturing interaction-aware dynamics while enabling parallel computation and avoiding iterative updates. Here, $A$ directly conditions the flow mapping $F(\mathbf{z}_0, t_0, t, A)$, allowing graph-structured interactions to modulate latent trajectory generation rather than only inference. To enforce trajectory consistency and integrate the interaction graph effectively, we further propose a *self-supervised trajectory generation* mechanism, which leverages both re-initialization and reversible trajectory constraints (Section 5.2). The following subsections describe each component in detail.

### 5.1. Initial Latent Condition Inference

Since exact posterior inference of the interaction graph $A$ and the initial latent state $\mathbf{z}_0$ is intractable (Kingma & Welling, 2013), we approximate them using variational distributions $q_\phi(A \mid X)$ and $q_\phi(\mathbf{z}_0 \mid X)$ as described below. During inference, missing observations are zero-imputed.

**Posterior of the interaction graph.** To robustly infer interaction structure under irregular missing observations, we perform graph inference over local temporal segments and aggregate segment-level posteriors into a global interac-

tion graph. Specifically, segment-level adjacency matrices $\{A^{(s)}\}_{s=1}^C$ are introduced during inference to capture locally reliable interactions and are aggregated into a single posterior $q_\phi(A \mid X)$, which defines the time-invariant interaction graph used by the generative model.

*(i) Segment-level representation.* Let $\{X^{(s)}\}_{s=1}^C$ denote $C$ consecutive local temporal segments of length $M = L/C$. We compute a segment-level representation by temporal averaging as $\bar{\mathbf{x}}^{(s)} = \frac{1}{M} \sum_{k=1}^M \mathbf{x}_{(s-1)M+k}$.

*(ii) Segment-level adjacency inference.* Variable interactions are inferred via self-attention (Zhou et al., 2023), with query and key vectors $\mathbf{q}_i^{(s)} = \bar{\mathbf{x}}_i^{(s)} W^Q$ and $\mathbf{k}_i^{(s)} = \bar{\mathbf{x}}_i^{(s)} W^K$ for variable $i$, where $\bar{x}_i^{(s)} \in \mathbb{R}$ denotes the segment-level average of variable $i$, and $W^Q, W^K \in \mathbb{R}^{1 \times D_x}$ are learnable linear projection matrices. The adjacency weight from variable $i$ to $j$ is computed as $a_{ij}^{(s)} = \frac{\exp(\mathbf{q}_i^{(s)}(\mathbf{k}_j^{(s)})^\top / \sqrt{D_x})}{\sum_{j'=1}^{D_x} \exp(\mathbf{q}_i^{(s)}(\mathbf{k}_{j'}^{(s)})^\top / \sqrt{D_x})}$, forming the *segment-level* adjacency matrix $A^{(s)} = [a_{ij}^{(s)}]$.

*(iii) Posterior aggregation.* Each $A^{(s)}$ parameterizes a Gaussian variational component $q_\phi(A^{(s)} \mid X^{(s)}) = \mathcal{N}(\mu_{A^{(s)}}, \sigma_{A^{(s)}})$, where $\mu_{A^{(s)}} = h_A(A^{(s)})$ and $\sigma_{A^{(s)}} = \mathrm{Softplus}(h_A(A^{(s)}))$, with $h_A(\cdot)$ denoting a shared feed-forward network. Gaussian samples are normalized via softmax to obtain valid adjacency matrices. The posterior over the full interaction graph is a weighted mixture of segment-level posteriors:

$$q_\phi(A \mid X) = \sum_{s=1}^C w_s \cdot q_\phi(A^{(s)} \mid X^{(s)}), \tag{4}$$

with mixture weights $w_s = \frac{D_{KL}(q_\phi(A^{(s)} | X^{(s)}) \| p(A))}{\sum_{j=1}^C D_{KL}(q_\phi(A^{(j)} | X^{(j)}) \| p(A))}$, where $p(A)$ denotes a standard Gaussian prior. These weights assign higher importance to segments that induce more informative posteriors, thereby mitigating the effect of irregular missingness.

**Posterior of the Initial Latent State.** At each time point $t_k$, the observation $\mathbf{x}_k$ and its mask $\mathbf{m}_k$ are concatenated and encoded into a latent representation $\mathbf{z}(t_k) = f_{\mathrm{enc}}(\mathbf{x}_k \mid \mathbf{m}_k)$. Here, the encoder outputs $z(t_k) \in \mathbb{R}^{D_x \times d_z}$, where $d_z$ denotes the latent dimension of each variable. To infer the initial latent state $\mathbf{z}_0$, we follow the IVP-VAE paradigm (Xiao et al., 2024) and treat each latent state $(\mathbf{z}(t_k), t_k)$ as an initial condition of the underlying continuous-time system. Specifically, for each time point $t_k$, we identify its corresponding temporal segment indexed by $s_k = \lceil k/M \rceil$ and integrate the associated segment-conditioned adjacency matrix $A^{(s_k)}$ to propagate $\mathbf{z}(t_k)$ backward to $t_0$ via the proposed GSNF:

$$\{\mathbf{z}_0^k\}_{k=1}^L = \left\{ F\big(\mathbf{z}(t_k), t_k, t_0, A^{(s_k)}\big) \right\}_{k=1}^L. \tag{5}$$

The segment-conditioned adjacency matrices $\{A^{(s)}\}_{s=1}^C$ are introduced only during inference to improve robustness under irregular missing observations. Each IVP-based estimate $\mathbf{z}_0^k$ parameterizes a diagonal Gaussian variational component $q_\phi(\mathbf{z}_0^k \mid X) = \mathcal{N}(\mu_{\mathbf{z}_0^k}, \sigma_{\mathbf{z}_0^k})$, where $\mu_{\mathbf{z}_0^k} = h_z(\mathbf{z}_0^k)$ and $\sigma_{\mathbf{z}_0^k} = \text{Softplus}(h_z(\mathbf{z}_0^k))$, with $h_z(\cdot)$ a shared feed-forward network. Finally, the posterior distribution over the initial latent state $\mathbf{z}_0$ is obtained from $\{\mathbf{z}_0^k\}_{k=1}^L$ of the sequence $X$:

$$q_\phi(\mathbf{z}_0 \mid X) = \frac{1}{L} \sum_{i=1}^L q_\phi(\mathbf{z}_0^i \mid X). \tag{6}$$

### 5.2. Latent Trajectory Generation

With the initial latent state $\mathbf{z}_0$ sampled from (6) and the interaction graph $A$ sampled from (4), $(\mathbf{z}_0, A)$ form the joint initial condition of the graph-structured flow. Latent trajectories are approximated using the GSNF as

$$\{\mathbf{z}(t_i)\}_{i=1}^L = \{F(\mathbf{z}_0, 0, t_i, A)\}_{i=1}^L. \tag{7}$$

However, one-step generation applies the learned interactions only once, offering limited supervision for refining the continuous dynamics and variable dependencies encoded in the flow and the interaction graph. Therefore, we introduce self-supervised regularization by generating two complementary auxiliary trajectories and enforcing trajectory-level constraints, as detailed below.

**Interaction-Aware Trajectory Generation (ITG).** Following the observations in (Mercatali et al., 2024), non-interacting components exhibit invariant behavior under perturbations, whereas components involved in graph-induced interactions show sensitivity to re-initialization. Building on this insight, GSNF constructs interaction-aware auxiliary trajectories by re-initializing the learned flow approximation along the original latent trajectory.

Specifically, for a re-initialization time $t_0^* \in (t_0, t_L)$, we define the corresponding state $\mathbf{z}_0^* = F(\mathbf{z}(t_0), t_0, t_0^*, A)$, and generate an auxiliary trajectory using the same parameterized GSNF, with $\mathbf{z}_0^*$ serving as the re-initial condition:

$$\mathbf{z}^*(t) = F(\mathbf{z}_0^*, t_0^*, t, A), \quad t \geq t_0^*. \tag{8}$$

The divergence between the original trajectory $\mathbf{z}(t)$ and the re-initialized auxiliary trajectory $\mathbf{z}^*(t)$ reflects the sensitivity of the learned flow to interaction-induced perturbations. To explicitly encourage such sensitivity, we introduce a margin-based trajectory-level regularizer:

$$\mathcal{L}_{\text{ITG}} = \frac{1}{L - k_0^* + 1} \max\left(\delta^2 - \sum_{i=k_0^*}^L \|\mathbf{z}(t_i) - \mathbf{z}^*(t_i)\|^2, 0\right), \tag{9}$$

where the summation starts from the re-initialization index $k_0^*$ for $t_i \geq t_0^*$, and $\delta > 0$ sets a separation margin.

In this work, the separation margin $\delta$ can be treated as a fixed hyperparameter, or alternatively chosen based on a theoretical lower-bound analysis provided in Theorem 5.1, without manual tuning.

**Theorem 5.1** (A Data-Dependent Lower Bound for the ITG Separation Margin). *Assume that the graph-conditioned interaction module in Eq.* (3) *admits the form* $\text{GCN}(\mathbf{z}(t_0), t_0, t, A) = \mathcal{A}\mathbf{z}(t_0)W$, *where* $\mathcal{A}$ *is the normalized adjacency matrix and* $W$ *is a trainable weight matrix. Let* $\mathbf{z}(t)$ *and* $\mathbf{z}^*(t)$ *denote the original and re-initialized latent trajectories, respectively. Then, for all discrete time points* $t_i > t_0^*$, *the cumulative trajectory divergence satisfies*

$$\sum_{i=k_0^*}^L \|\mathbf{z}^*(t_i) - \mathbf{z}(t_i)\| \geq \max\left\{0, (L - k_0^* + 1)(\eta - \Delta_{\text{in}})\right\}, \tag{10}$$

*where* $\Delta_{\text{in}} = \|\mathbf{z}_0^* - \mathbf{z}_0\|$, $\eta = \sigma_{\min}(\mathcal{A})\,\sigma_{\min}(W)\,\Delta_{\text{in}}$, *and* $k_0^*$ *denotes the index of the first time point* $t_i \geq t_0^*$, $\sigma_{\min}(\cdot)$ *denotes the smallest singular value of a matrix.*

This bound characterizes the minimal divergence induced by the interaction structure when re-initialization perturbations are amplified by graph-conditioned dynamics. The proof is provided in Appendix B.

**Reverse-time Trajectory Generation (RTG).** Leveraging the invertibility of GSNF guaranteed by Theorem 4.1, we generate an additional self-supervised signal by reversing the learned latent trajectory. Starting from the final latent state $\mathbf{z}(t_L)$ obtained in Eq.(7), we apply the learned flow in reverse time to obtain a reversed latent trajectory $\{\tilde{\mathbf{z}}(t_k)\}_{k=1}^L$:

$$\{\tilde{\mathbf{z}}(t_i)\}_{i=1}^L = \{F(\mathbf{z}(t_L), t_L, t_i, A)\}_{i=1}^L. \tag{11}$$

The reversed trajectory is decoded and compared to the input sequence using the reconstruction loss:

$$\mathcal{L}_{\text{RTG}} = \frac{1}{L} \sum_{i=1}^L \|f_{\text{dec}}(\tilde{\mathbf{z}}(t_i)) - \mathbf{x}_i\|^2. \tag{12}$$

This reverse-time regularization encourages forward–backward consistency of the learned flow.

### 5.3. Training

Our model is a VAE-based framework augmented with a generative flow to capture latent interactions and ensure temporally consistent dynamics. The overall objective combines variational inference with supervised classification, and trajectory-level self-supervision:

$$\mathcal{L} = \mathcal{L}_{\text{VAE}} + \alpha\mathcal{L}_{\text{CE}} + \beta\,\mathcal{L}_{\text{ITG}} + \gamma\,\mathcal{L}_{\text{RTG}}, \tag{13}$$

where $L_{\mathrm{VAE}}$ is the evidence lower bound (ELBO) (Kingma & Welling, 2013),

$$
\begin{aligned}
\mathcal{L}_{\mathrm{VAE}}(\phi, \theta) = \; & \mathbb{E}_{\substack{z_0 \sim q_\phi(z_0|X) \\ A \sim q_\phi(A|X)}} \big[ \log p_\theta(X|z_0, A) \big] \\
& - \frac{1}{L} \sum_{i=1}^{L} D_{\mathrm{KL}} \big( q_\phi(z_0^i|X) \,\|\, p(z_0) \big) \\
& - \frac{1}{C} \sum_{s=1}^{C} D_{\mathrm{KL}} \big( q_\phi(A^{(s)}|X^{(s)}) \,\|\, p(A) \big).
\end{aligned}
\tag{14}
$$

and $\mathcal{L}_{\mathrm{CE}}$ is the cross-entropy loss for classification:

$$
\mathcal{L}_{\mathrm{CE}} = - \sum_y p(y) \log p_\lambda(y \mid \mathbf{z}_0).
\tag{15}
$$

The trajectory-level losses $\mathcal{L}_{\mathrm{ITG}}$ and $\mathcal{L}_{\mathrm{RTG}}$ act as regularizers complementing the main objective. $\mathcal{L}_{\mathrm{ITG}}$ enforces divergence between original and re-initialized trajectories to capture interactions, while $\mathcal{L}_{\mathrm{RTG}}$ promotes temporal consistency via reverse-time reconstruction. The relative weights $\alpha$, $\beta$, and $\gamma$ are reported in Appendix E.1, and the full training procedure is summarized in Algorithm 1.

**ITG Training Variants.** For the ITG loss, we consider two variants for defining the minimum separation margin: (i) a fixed hyperparameter $\delta$; (ii) a theoretically derived margin $\delta_{\mathrm{lb}}$ set according to the lower bound in Theorem 5.1. The latter requires no manual tuning, as the margin is automatically adapted during training along with the evolving model parameters.

## 6. Experiments

### 6.1. Datasets and Baselines

We evaluated GSNF on five datasets for irregular time series classification: PhysioNet12 (Silva et al., 2012), P12 (Goldberger et al., 2000), P19 (Reyna et al., 2020), MIMIC-IV (Johnson et al., 2020), eICU (Pollard et al., 2018). Detailed experimental settings are provided in Appendix E.

We compare with representative baselines from three major categories: (i) continuous-time models, including GRU-D (Che et al., 2018), ODE-RNN (Rubanova et al., 2019), NeuralFlow (Biloš et al., 2021), IVP-VAE (Xiao et al., 2024), DualDynamics (Oh et al., 2025b) and FlowPath (Oh et al., 2026); (ii) graph-based models, including Raindrop (Zhang et al., 2022), GNeuralFlow (Mercatali et al., 2024) and Hi-Patch (Luo et al., 2025); (iii) other strong baselines, including mTAN (Shukla & Marlin, 2021), ViTST (Li et al., 2023), Warpformer (Zhang et al., 2023), and TimeCHEAT (Liu et al., 2025).

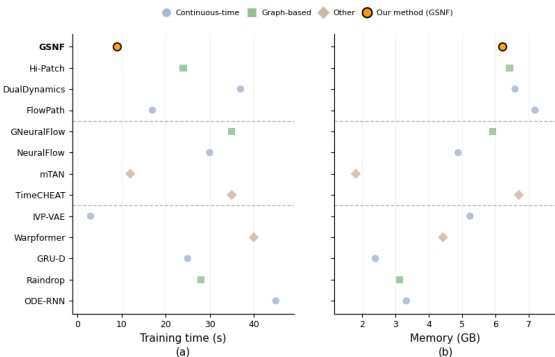

*Figure 3.* Computational efficiency on the PhysioNet12 dataset. (a) Training time and (b) peak GPU memory usage. Methods are ordered from top to bottom by decreasing AUPRC, with dashed lines indicating three performance tiers. GSNF achieves the highest AUPRC and attains the lowest training time and memory usage within the top performance tier. Different marker shapes denote different categories of methods.

### 6.2. Overall Comparisons

Overall results are summarized in Table 1, with variability shown in Fig. 8. Across all five datasets, the top two models in terms of AUROC and AUPRC are both variants of our method GSNF, indicating stable performance across datasets with diverse missing rates, label imbalance, and feature dimensionality.

Notably, GSNF achieves consistently strong improvements in AUPRC, which is more indicative than AUROC in highly imbalanced tasks. The largest gain is observed on P19, where GSNF outperforms the second-best method by 1.2 AUPRC points. Since P19 has the lowest positive rate among all evaluated datasets, this result suggests that incorporating inter-variable interactions is especially beneficial under severe class imbalance, enabling the model to better identify rare but clinically critical positive cases.

In addition, the variant of GSNF with the theoretically derived margin, GSNF($\delta_{\mathrm{lb}}$), outperforms the best manually tuned variant GSNF($\delta$) on four out of five datasets (PhysioNet12, P12, P19, and MIMIC-IV), and achieves comparable performance on eICU. This observation indicates that the theoretically derived margin $\delta_{\mathrm{lb}}$ provides a principled alternative to dataset-specific hyperparameter tuning and yields robust performance in practice. As a result, $\delta_{\mathrm{lb}}$ offers a reliable and practical mechanism for regulating trajectory separation without incurring additional tuning overhead.

Fig. 3 further examines computational efficiency. Within the top AUPRC tier, GSNF achieves the lowest training time. This efficiency is largely attributed to the IVP-VAE backbone, which exhibits the lowest training time among all the baselines. Notably, despite being the fastest baseline, IVP-VAE lies in the third performance tier; GSNF incor-

| Method | PhysioNet12 | | P12 | | P19 | | MIMIC-IV | | eICU | |
|---|---|---|---|---|---|---|---|---|---|---|
| | AUROC | AUPRC | AUROC | AUPRC | AUROC | AUPRC | AUROC | AUPRC | AUROC | AUPRC |
| GRU-D | 79.1±6.9 | 42.7±7.2 | 81.9±2.1 | 46.1±4.7 | 83.7±1.5 | 46.9±2.1 | 82.2±1.8 | 48.3±2.1 | 84.6±1.5 | 49.6±2.4 |
| ODE-RNN | 80.8±0.4 | 33.7±4.1 | 81.5±0.5 | 42.3±0.7 | 81.4±0.8 | 45.7±1.0 | 81.0±0.6 | 47.3±0.7 | 83.3±0.9 | 50.3±1.5 |
| NeuralFlow | 80.9±0.1 | 51.5±1.8 | 81.3±0.1 | 51.5±2.0 | 84.1±0.7 | 52.1±3.3 | 79.7±0.4 | 51.5±1.2 | 83.1±0.8 | 50.7±1.8 |
| IVP-VAE | 81.1±2.3 | 46.2±2.3 | 81.8±0.2 | 53.5±1.5 | 85.6±1.2 | 53.7±2.7 | 81.8±0.5 | 52.7±1.4 | 83.6±1.7 | 53.2±2.9 |
| DualDynamics | 86.1±0.1 | 55.3±1.5 | 86.5±0.2 | 53.2±0.8 | 90.7±0.4 | 56.7±1.4 | 84.4±0.4 | 54.7±0.9 | 84.9±1.1 | 54.2±1.5 |
| FlowPath | 85.3±0.2 | 55.3±1.7 | 86.6±0.3 | 54.3±0.7 | 88.4±0.7 | 56.4±1.3 | 84.4±0.7 | 53.2±0.8 | 85.1±1.3 | 53.8±1.4 |
| Raindrop | 81.2±0.9 | 37.3±2.0 | 82.8±1.7 | 39.4±2.4 | 87.0±2.3 | 51.8±5.5 | 79.8±1.3 | 35.2±1.1 | 84.6±2.1 | 55.1±2.7 |
| GNeuralFlow | 84.5±0.8 | 53.7±2.4 | 85.5±0.8 | 55.5±1.8 | 88.4±0.7 | 56.4±1.3 | 83.6±0.7 | 53.1±0.8 | 85.1±0.7 | 54.5±2.6 |
| Hi-Patch | 86.4±0.9 | 56.5±1.4 | 86.5±0.6 | 53.3±0.9 | 88.4±1.1 | 56.2±3.3 | 84.9±0.2 | 54.2±1.0 | 84.5±0.7 | 55.3±1.3 |
| mTAN | 85.8±1.3 | 50.4±1.0 | 84.2±0.8 | 52.5±1.3 | 84.0±1.3 | 50.6±2.0 | 83.8±0.3 | 46.6±0.5 | 80.9±2.4 | 48.1±3.2 |
| ViTST | 81.3±1.9 | 37.4±2.9 | 86.3±0.1 | 50.8±1.5 | 88.3±2.0 | 53.1±3.4 | 81.8±0.3 | 39.6±1.3 | 80.5±1.3 | 47.7±2.4 |
| Warpformer | 83.4±0.7 | 43.5±2.3 | 85.4±0.5 | 50.4±1.5 | 87.2±1.8 | 52.7±2.4 | 84.6±0.3 | 46.6±0.9 | 84.8±0.4 | 53.7±1.1 |
| TimeCHEAT | 84.5±0.7 | 46.3±1.5 | 84.6±0.7 | 48.2±1.9 | 89.1±1.4 | 55.2±1.9 | 83.1±0.3 | 51.2±0.9 | 83.9±1.4 | 54.4±1.5 |
| GSNF($\delta$) | 86.4±1.0 | 56.8±1.0 | 86.6±0.1 | 54.6±1.4 | 90.7±0.6 | 57.4±2.3 | 84.9±0.5 | 55.3±1.3 | **85.2±1.4** | 56.1±1.0 |
| GSNF($\delta_{lb}$) | **86.7±0.6** | **56.9±0.7** | **87.5±0.3** | **56.3±0.4** | **91.2±0.5** | **57.9±2.4** | **85.3±0.5** | **55.5±1.5** | 85.2±0.9 | **56.1±1.1** |

*Table 1.* Comparison of GSNF and baseline methods on classification tasks across five datasets. GSNF variants achieve the top performance across datasets, with the theoretically derived variant GSNF($\delta_{lb}$) performing better in most cases.

| Method | PhysioNet12 | | P12 | | P19 | | MIMIC-IV | | eICU | |
|---|---|---|---|---|---|---|---|---|---|---|
| | AUROC | AUPRC | AUROC | AUPRC | AUROC | AUPRC | AUROC | AUPRC | AUROC | AUPRC |
| GSNF | **86.7** | **56.9** | **87.5** | **56.3** | **91.2** | **57.9** | **85.3** | **55.5** | **85.2** | **56.1** |
| -w/o ITG | 84.4 | 51.3 | 84.7 | 51.2 | 87.9 | 54.4 | 85.1 | 52.3 | 84.2 | 52.1 |
| -w/o RTG | 85.5 | 53.7 | 85.5 | 53.0 | 89.2 | 55.7 | 84.5 | 53.9 | 84.0 | 53.7 |
| -w/o RTG&ITG | 84.1 | 48.9 | 83.9 | 50.1 | 86.1 | 53.8 | 83.1 | 51.9 | 83.5 | 51.4 |
| -w/o graph | 82.7 | 46.4 | 82.7 | 48.2 | 84.8 | 52.9 | 81.4 | 50.2 | 81.6 | 49.8 |

*Table 2.* Ablation study. The -w/o graph variant yields the largest drop, especially in AUPRC, followed by -w/o ITG+RTG; -w/o ITG degrades performance more than -w/o RTG.

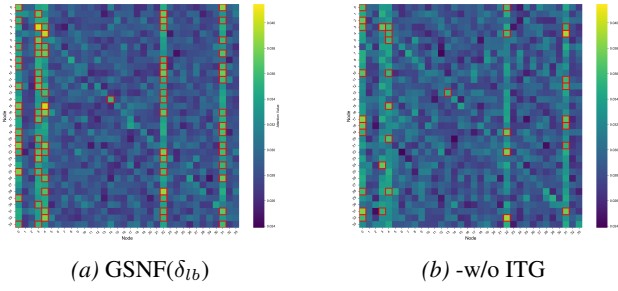

*(a)* GSNF($\delta_{lb}$)  *(b)* -w/o ITG

*Figure 4.* Interaction graphs with and without ITG on the P19 dataset. Attention weights are normalized by softmax, and edges highlighted by red boxes denote attention values above 0.035. With ITG, the interaction graph contains substantially more salient connections (78 edges) than without ITG (34 edges), with the additional connections primarily concentrated around key clinical variables. (0: HR, 3: SBP, 4: MAP, 22: lactate, 31: WBC).

porating interaction modeling resolves this limitation by elevating GSNF to the top AUPRC tier, while retaining the second-lowest training time overall. Meanwhile, peak GPU memory generally increases with higher AUPRC across methods. For clarity, only one GSNF point is shown in Fig. 3, as the two GSNF variants have comparable training time and memory usage due to the negligible overhead of the lower-bound computation. Detailed numerical results are provided in Appendix D.1.

## 6.3. Ablation Study

We construct four ablation variants of GSNF by removing ITG, RTG, both ITG and RTG, or the interaction graph. Table 2 presents the corresponding ablation results across five datasets, showing consistent performance degradation when any component is removed.

Among all components, removing the interaction graph results in the largest performance degradation across datasets, especially in AUPRC, highlighting the necessity of explicitly modeling inter-variable interactions. Removing both

ITG and RTG leads to the second-largest performance drop, indicating that these two trajectory-level supervision mechanisms jointly contribute to stabilizing and refining interaction learning. When considered individually, removing ITG causes a larger performance decrease than removing RTG.

Beyond numerical performance, Fig. 4 provides a qualitative view of how ITG influences interaction learning. Attention weights are normalized by softmax, and red boxes indicate edges whose attention values exceed 0.035. In both GSNF and its w/o ITG ablation, high-attention connections are primarily associated with a small set of clinically important variables (0: HR, heart rate; 3: SBP, systolic blood pressure; 4: MAP, mean arterial pressure; 22: lactate; 31:

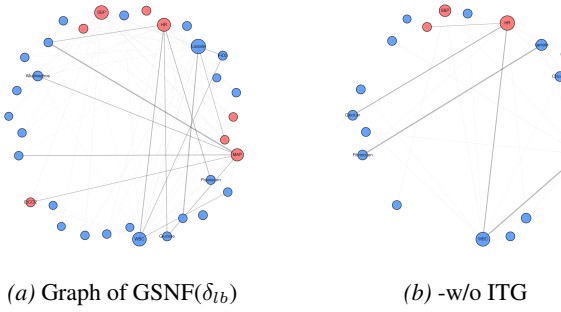

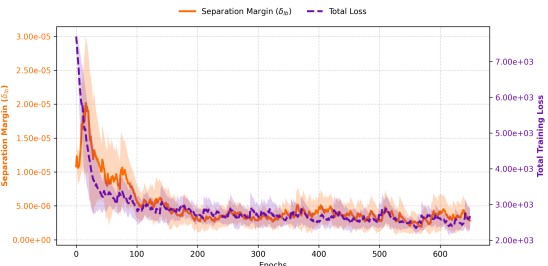

*(a)* Graph of GSNF($\delta_{lb}$)     *(b)* -w/o ITG

*Figure 5.* Visualization of learned interaction graph on P19. Node size reflects weighted degree, node color denotes variable type (red: vital signs; blue: laboratory variables; left: 8/25; right: 5/15). Only the top 20% strongest interactions are shown, with labels displayed for the most connected nodes, highlighting the structural differences induced by ITG.

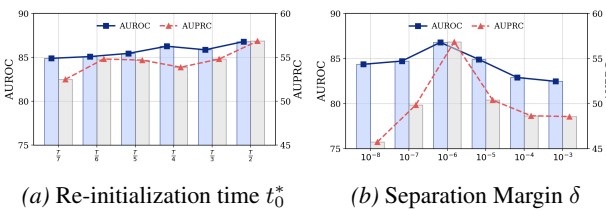

*(a)* Re-initialization time $t_0^*$     *(b)* Separation Margin $\delta$

*Figure 6.* Parameter sensitivity analysis of GSNF on PhysioNet12. Performance peaks at an intermediate $t_0^*$ and depends on $\delta$ mainly at the order-of-magnitude scale, with an optimum around $10^{-6}$.

WBC, white blood cell count), forming visually prominent vertical regions. When ITG is enabled, the number of edges highlighted by red boxes increases from 34 to 78, with the additional connections remaining concentrated around these variables. This suggests that ITG reinforces coordinated interactions rather than introducing diffuse or noisy edges.

Further, Fig. 5 complements this analysis by visualizing the learned interaction graphs. The interaction graph learned by GSNF exhibits stronger and denser connectivity, with HR, SBP, MAP, lactate, and WBC acting as clear hub nodes around which high-weight edges concentrate to form a coherent interaction backbone. In contrast, the w/o ITG ablation produces few strong edges, with limited connectivity and substantially reduced structural complexity. Most nodes remain weakly connected or isolated, leading to a sparse graph that captures only a small subset of interactions.

### 6.4. Parameters Sensitivity

We analyze the sensitivity of ITG to the re-initialization time and the tuned margin $\delta$ on PhysioNet12, as shown in Figure 6. Cross-dataset tuning results and the theoretical lower-bound $\delta_{lb}$ are further provided in Appendix D.2, while Fig. 7 reports its behavior during training as a non-tunable reference.

*Figure 7.* Training dynamics of the theoretically derived separation margin $\delta_{lb}$ and total loss on PhysioNet12. The theoretically derived margin $\delta_{lb}$ maintains a comparable scale to the manually selected margin $\delta$, on the order of $10^{-6}$, across training.

**Re-initialization time $t_0^*$.** Figure 6(a) shows that GSNF achieves the best performance when the re-initialization time $t_0^*$ is set to an intermediate point around $L/2$. When $t_0^*$ is chosen too early, the re-initialized trajectories remain similar to the original ones, resulting in limited divergence and weakened ITG supervision. In contrast, setting $t_0^*$ too late reduces the effective evolution window, restricting interaction exposure and leading to degraded performance.

**Manually Selected Separation Margin $\delta$.** Figure 6(b) shows that while performance varies across orders of magnitude of $\delta$, it remains relatively stable within each order, with the best results attained around $\delta = 10^{-6}$. This indicates that GSNF is robust to fine-grained variations of $\delta$ and does not require precise tuning. This is because overly large margins induce excessive trajectory divergence, whereas overly small margins provide insufficient separation of interaction-induced differences, reflecting a trade-off between interaction exposure and dynamical fidelity.

**Theoretically Derived Separation Margin $\delta_{lb}$.** As shown in Fig. 7, $\delta_{lb}$ increases during the early stages of training and gradually converges to a stable value as training proceeds. This behavior reflects the data-dependent nature of the theoretical bound, which adjusts with the training process rather than remaining fixed. The converged value lies within the same order of magnitude as the manually selected separation margin (around $10^{-6}$ on PhysioNet12), indicating that the theoretical bound captures an appropriate scale of trajectory divergence without manual tuning. Compared with a fixed manually selected separation margin $\delta$, the data-dependent theoretically derived margin $\delta_{lb}$ enables more flexible trajectory separation during training and leads to improved classification performance.

## 7. Conclusion

We propose GSNF, a one-step Graph-Structured Neural Flow that integrates an interaction graph into a parallel flow

formulation for irregularly sampled multivariate time series. GSNF introduces trajectory-level self-supervision via ITG and RTG, improving interaction learning without iterative solvers. Experiments on five real-world datasets demonstrate that GSNF variants consistently achieve state-of-the-art performance, with GSNF($\delta_{1b}$), which uses a theoretically derived separation margin, performing best in most cases. Among top-performing methods, GSNF achieves the lowest training time and peak GPU memory.

Despite its effectiveness, GSNF may face challenges in modeling extremely long sequences and highly complex dynamics. Moreover, the current formulation assumes static interaction graphs at inference time, limiting its ability to capture rapidly evolving dependencies in non-stationary settings. Future work will extend GSNF to time-varying interaction graphs and improve scalability in large-scale non-stationary systems.

## Acknowledgements

This work was supported in part by the Zhejiang Provincial Natural Science Foundation under Grant LMS25F030011, in part by the National Natural Science Foundation of China under Grant 62372146, in part by the Zhejiang Province Key R&D Program Project under Grant No. 2025C01023, and in part by the Zhejiang Provincial Key Laboratory for Sensitive Data Security Protection and Confidentiality Management under Grant No. 2024E10048.

## Impact Statement

This paper presents work whose goal is to advance the field of machine learning. There are many potential societal consequences of our work, none of which we feel must be specifically highlighted here.

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

## A. Proof of Theorem 4.1 (Invertibility of Graph-Structured Neural Flows)

*Proof.* Let $F(\mathbf{z}(t_0), t_0, t, A)$ denote the GSNF defined in Eq. (3). We assume that the interaction function $g(\cdot, t_0, t, A)$ is contractive with Lipschitz constant $L_g < 1$, which is ensured by applying spectral normalization to all linear layers in both the MLP and GCN.

To establish invertibility, we show that $F(\cdot, t_0, t, A)$ is bi-Lipschitz. For any two latent states $\mathbf{z}_1(t_0)$ and $\mathbf{z}_2(t_0)$, we have

$$
\begin{aligned}
\|F(\mathbf{z}_1(t_0), t_0, t, A) - F(\mathbf{z}_2(t_0), t_0, t, A)\| &= \big\|\mathbf{z}_1(t_0) - \mathbf{z}_2(t_0) + \varphi(t - t_0)\big(g(\mathbf{z}_1(t_0), t_0, t, A) - g(\mathbf{z}_2(t_0), t_0, t, A)\big)\big\| \\
&\le \|\mathbf{z}_1(t_0) - \mathbf{z}_2(t_0)\| + \varphi(t - t_0)\|g(\mathbf{z}_1(t_0), t_0, t, A) - g(\mathbf{z}_2(t_0), t_0, t, A)\| \\
&\le (1 + \varphi(t - t_0)L_g)\|\mathbf{z}_1(t_0) - \mathbf{z}_2(t_0)\|.
\end{aligned}
\tag{16}
$$

On the other hand, by the reverse triangle inequality,

$$
\begin{aligned}
\|F(\mathbf{z}_1(t_0), t_0, t, A) - F(\mathbf{z}_2(t_0), t_0, t, A)\| &\ge \|\mathbf{z}_1(t_0) - \mathbf{z}_2(t_0)\| - \varphi(t - t_0)\|g(\mathbf{z}_1(t_0), t_0, t, A) - g(\mathbf{z}_2(t_0), t_0, t, A)\| \\
&\ge (1 - \varphi(t - t_0)L_g)\|\mathbf{z}_1(t_0) - \mathbf{z}_2(t_0)\|.
\end{aligned}
\tag{17}
$$

Since $\varphi(t - t_0) \in [0, 1)$ and $L_g < 1$, we have $1 - \varphi(t - t_0)L_g > 0$. Together with the upper and lower bounds derived above, this implies that $F(\cdot, t_0, t, A)$ satisfies

$$
0 < \big(1 - \varphi(t - t_0)L_g\big)\|\mathbf{z}_1(t_0) - \mathbf{z}_2(t_0)\| \le \|F(\mathbf{z}_1(t_0), t_0, t, A) - F(\mathbf{z}_2(t_0), t_0, t, A)\| \le \big(1 + \varphi(t - t_0)L_g\big)\|\mathbf{z}_1(t_0) - \mathbf{z}_2(t_0)\|,
\tag{18}
$$

and is therefore bi-Lipschitz.

By standard properties of bi-Lipschitz mappings, its inverse $F^{-1}$ is Lipschitz continuous with Lipschitz constant bounded by $1/(1 - \varphi(t - t_0)L_g)$. $\qquad\square$

## B. Proof of Theorem 5.1 (A Data-Dependent Lower Bound for the ITG Separation Margin)

*Proof.* Recall that the GSNF adopts a residual flow parameterization

$$
F(\mathbf{z}(t_0), t_0, t, A) = \mathbf{z}(t_0) + \varphi(t - t_0)\, g(\mathbf{z}(t_0), t_0, t, A).
\tag{19}
$$

Let $\mathbf{z}(t)$ and $\mathbf{z}^*(t)$ denote the original and re-initialized latent trajectories with initial states $\mathbf{z}_0$ and $\mathbf{z}_0^*$, respectively. For $t \ge t_0^*$, define $\delta(t) := \mathbf{z}^*(t) - \mathbf{z}(t)$. Then

$$
\begin{aligned}
\delta(t) &= F(\mathbf{z}_0^*, t_0^*, t, A) - F(\mathbf{z}_0, t_0, t, A) \\
&= (\mathbf{z}_0^* - \mathbf{z}_0) + \varphi(t - t_0^*)\, g(\mathbf{z}_0^*, t_0^*, t, A) - \varphi(t - t_0)\, g(\mathbf{z}_0, t_0, t, A).
\end{aligned}
\tag{20}
$$

Taking norms and applying the reverse triangle inequality yields

$$
\|\delta(t)\| \ge \|\varphi(t - t_0^*)\, g(\mathbf{z}_0^*, t_0^*, t, A) - \varphi(t - t_0)\, g(\mathbf{z}_0, t_0, t, A)\| - \|\mathbf{z}_0^* - \mathbf{z}_0\|.
\tag{21}
$$

Denote $\Delta_{\mathrm{in}} := \|\mathbf{z}_0^* - \mathbf{z}_0\|$.

We now lower-bound the residual term. Recall that

$$
g(\mathbf{z}(t_0), t_0, t, A) = \mathrm{MLP}(\mathbf{z}(t_0), t_0, t) \odot \mathrm{GCN}(\mathbf{z}(t_0), t_0, t, A),
$$

and assume the linear form $\mathrm{GCN}(\mathbf{z}(t_0), t_0, t, A) = \mathcal{A}\,\mathbf{z}(t_0)\,W$, where $\mathcal{A}$ is the normalized adjacency matrix. All linear layers in both the MLP and the GCN are spectrally normalized, so that the MLP defines a Lipschitz-bounded element-wise modulation.

Then, for $t \ge t_0^*$, we define

$$
\eta(t) := \|\varphi(t - t_0^*)\, g(\mathbf{z}_0^*, t_0^*, t, A) - \varphi(t - t_0)\, g(\mathbf{z}_0, t_0, t, A)\|.
\tag{22}
$$

By the spectral properties of the spectrally normalized MLP and the linear GCN, $\eta(t)$ admits the lower bound

$$\eta(t) \geq \varphi(t - t_0^*) \sigma_{\min}(\mathcal{A}) \sigma_{\min}(W) \|\mathbf{z}_0^* - \mathbf{z}_0\|. \tag{23}$$

Substituting into Eq. (21) gives, for all $t \geq t_0^*$,

$$\|\delta(t)\| \geq \eta(t) - \Delta_{\text{in}} \geq \Big(\sigma_{\min}(\mathcal{A})\sigma_{\min}(W) - 1\Big)\Delta_{\text{in}}. \tag{24}$$

Summing over discrete trajectory points $\{t_i\}_{i=k_0^*}^{L}$ yields

$$\sum_{i=k_0^*}^{L} \|\mathbf{z}^*(t_i) - \mathbf{z}(t_i)\| \geq \sum_{i=k_0^*}^{L} \big(\eta(t_i) - \Delta_{\text{in}}\big) \geq (L - k_0^* + 1)\big(\eta - \Delta_{\text{in}}\big), \tag{25}$$

where we define $\eta := \sigma_{\min}(\mathcal{A})\sigma_{\min}(W)\Delta_{\text{in}}$. Since the left-hand side is nonnegative, we finally obtain

$$\sum_{i=k_0^*}^{L} \|\mathbf{z}^*(t_i) - \mathbf{z}(t_i)\| \geq \max\Big\{0, (L - k_0^* + 1)\big(\eta - \Delta_{\text{in}}\big)\Big\}, \tag{26}$$

which completes the proof. $\square$

## C. Training Algorithm

We present the complete training procedure of GSNF in Algorithm 1, which summarizes graph inference, initial state inference, forward trajectory generation, and trajectory-level regularization via ITG and RTG.

---

**Algorithm 1** Training Procedure for GSNF

---

1: **Input:** Dataset $\mathcal{X}$; Labels $\mathcal{Y}$; Hyperparameters $\alpha, \beta, \gamma$; Re-initialization time $t_0^*$.
2: **Output:** Optimized parameters
3: **while** not converged **do**
4:      Sample batch $(X, y)$ from $\mathcal{X}$.
5:      *// Stage 1. Interaction Graph Inference*:
6:      Compute segment-level adjacencies $A^{(s)}$ via self-attention;
7:      Aggregate global posterior $q_\phi(A|X) \leftarrow \sum w_s q_\phi(A^{(s)}|X^{(s)})$ (Eq. 4);
8:      Sample interaction graph $A \sim q_\phi(A|X)$;
9:      *// Stage 2. Initial Latent State Inference*:
10:     Back-propagate observed states to $t_0$ via Inverse GSNF: $z_0^k \leftarrow F^{-1}(z(t_k), \dots)$;
11:     Aggregate posterior $q_\phi(z_0|X)$ and sample $z_0$ (Eq. 6);
12:     *// Stage 3. Generation & Classification*:
13:     Evolve trajectory $z(t) \leftarrow F(z_0, t_0, t, A)$ (Eq. 7);
14:     Compute prediction $\hat{y}$ and losses $\mathcal{L}_{VAE}, \mathcal{L}_{CE}$;
15:     *// Stage 4. Trajectory-Level Self-Supervised Regularization*:
16:     **ITG:** Re-initialize at $t_0^*$ to get $z^*(t)$
17:     Determine separation margin $\delta$:
18:       Option 1: Set $\delta$ (Manually Tuned Hyperparameter);
19:       Option 2: Compute $\delta \leftarrow \delta_{lb}$ (Theoretical Lower Bound in Eq.10);
20:     Compute $\mathcal{L}_{ITG}$ using $\delta$ (Eq. 9);
21:     **RTG:** Reverse flow from $z(t_L)$ to get $\tilde{z}(t)$. Compute $\mathcal{L}_{RTG}$ (Eq. 12);
22:     *// Stage 5. Update:*
23:     Minimize total loss $\mathcal{L}$ (Eq. 13).
24: **end while**

---

Among the components in Algorithm 1, the ITG regularization step (Stage 4) relies on a separation margin $\delta$, whose choice is critical for effective trajectory separation. We consider two strategies: a manually tuned hyperparameter and a

data-dependent theoretical lower bound $\delta_{lb}$ derived in Eq. 10. The latter provides a parameter-free alternative for deployment across datasets.

## D. Comprehensive Experiments

### D.1. Memory Usage and Training Time

Table 3 presents the classification performance and computational efficiency of the proposed GSNF($\delta_{lb}$) variant compared to state-of-the-art baselines on the PhysioNet12 dataset. In terms of training speed, GSNF requires 9 seconds per epoch, making it the fastest among the top-performing models. Compared to high-accuracy baselines like DualDynamics (37s/epoch) and FlowPath (17s/epoch), GSNF achieves a speedup of approximately $2\times$ to $4\times$. This efficiency is attributed to the one-step flow architecture, which avoids the iterative integration required by ODE-based solvers. Regarding memory usage, GSNF consumes 6349 MB, which is comparable to other graph-based approaches.

| Method | AUROC | AUPRC | Time (s) | Memory (MB) |
|---|---|---|---|---|
| GRU-D | 79.1 | 42.7 | 25 | 2456 |
| ODE-RNN | 80.8 | 33.7 | 45 | 3407 |
| NeuralFlow | 80.9 | 51.5 | 30 | 4999 |
| IVP-VAE | 81.1 | 46.2 | **3** | 5363 |
| DualDynamics | 86.1 | 55.3 | 37 | 6743 |
| FlowPath | 85.3 | 55.3 | 17 | 7356 |
| RainDrop | 81.2 | 37.3 | 28 | 3200 |
| GraphNeuralFlow | 84.5 | 53.7 | 35 | 6056 |
| Hi-Patch | 86.4 | 56.5 | 24 | 6577 |
| mTAN | 85.8 | 50.4 | 12 | **1857** |
| Warpformer | 83.4 | 43.5 | 40 | 4532 |
| TimeCHEAT | 84.5 | 46.3 | 35 | 6859 |
| ViTST | 81.3 | 37.4 | 59 | 8982 |
| **GSNF (Ours)** | **86.7** | **56.9** | 9 | 6349 |

*Table 3.* Performance and efficiency comparison on PhysioNet12. **Bold** and underlined mark the best and second-best results, respectively.

### D.2. Cross-Dataset Tuning and Data-Dependent Behavior of Separation Margin

We study the behavior of the separation margin from two complementary perspectives: (i) its dataset-dependent tuning $\delta$ behavior under grid search, and (ii) its comparison with the proposed theoretical lower bound $\delta_{lb}$. Both aspects are evaluated on five datasets.

We first conduct a unified grid search over the candidate set $\{10^{-3}, 10^{-4}, 10^{-5}, 10^{-6}, 10^{-7}, 10^{-8}\}$ to identify the best-performing $\delta$ for each dataset. The results are reported in Table 4, where bold values indicate the selected optimal hyperparameters. We observe that the optimal $\delta$ varies across datasets, indicating strong data dependence.

| | PhysioNet12 | | P12 | | P19 | | MIMIC-IV | | eICU | |
|---|---|---|---|---|---|---|---|---|---|---|
| $\delta$ | AUROC | AUPRC | AUROC | AUPRC | AUROC | AUPRC | AUROC | AUPRC | AUROC | AUPRC |
| $10^{-3}$ | 82.46 | 48.56 | 85.02 | 53.67 | 89.74 | 55.91 | 83.54 | 52.01 | 83.75 | 55.02 |
| $10^{-4}$ | 82.88 | 48.64 | 85.75 | 54.01 | 90.13 | 56.45 | 83.24 | 52.86 | 83.87 | 54.92 |
| $10^{-5}$ | 84.89 | 50.38 | **86.24** | **54.55** | **90.76** | **57.32** | 83.56 | 52.61 | **84.05** | **55.48** |
| $10^{-6}$ | **86.78** | **56.84** | 84.64 | 52.98 | 90.38 | 56.83 | 84.01 | 53.34 | 83.14 | 54.37 |
| $10^{-7}$ | 84.71 | 49.84 | 83.21 | 51.77 | 89.97 | 56.47 | **84.54** | **54.87** | 82.87 | 53.15 |
| $10^{-8}$ | 84.35 | 45.73 | 82.54 | 51.54 | 89.51 | 55.92 | 83.78 | 54.63 | 82.55 | 52.86 |

*Table 4.* Cross-dataset tuning results of separation margin $\delta$, where bold values indicate the selected best-performing $\delta$ for each dataset.

In contrast, we further evaluate the proposed theoretical lower bound $\delta_{lb}$, derived in Eq. 10, which eliminates the need for dataset-specific tuning. Table 5 compares manually tuned $\delta$ with $\delta_{lb}$ across all datasets, showing that the theoretical bound provides a consistent, parameter-free scaling reference.

|  | PhysioNet12 | P12 | P19 | MIMIC-IV | eICU |
|---|---|---|---|---|---|
| $\delta$ | 1.00 | 10.00 | 10.00 | 0.10 | 10.00 |
| $\delta_{lb}$ | 3.51 | 22.54 | 15.76 | 0.47 | 29.82 |

*Table 5.* Comparison of $\delta$ and $\delta_{\text{lb}}$ across datasets ($\times 10^{-6}$).

### D.3. Main Results Visualization

Fig. 8 visualizes performance variability across five independent runs, complementing the main results in Table 1. The error bars, representing standard deviation, indicate that GSNF maintains consistently low variance across all datasets, demonstrating superior stability compared to baselines like Raindrop or FlowPath which exhibit higher volatility in certain metrics. Furthermore, the theoretical variant GSNF($\delta_{lb}$) shows comparable stability to the manually tuned GSNF($\delta$), confirming that using the calculated lower bound yields robust results without compromising training stability.

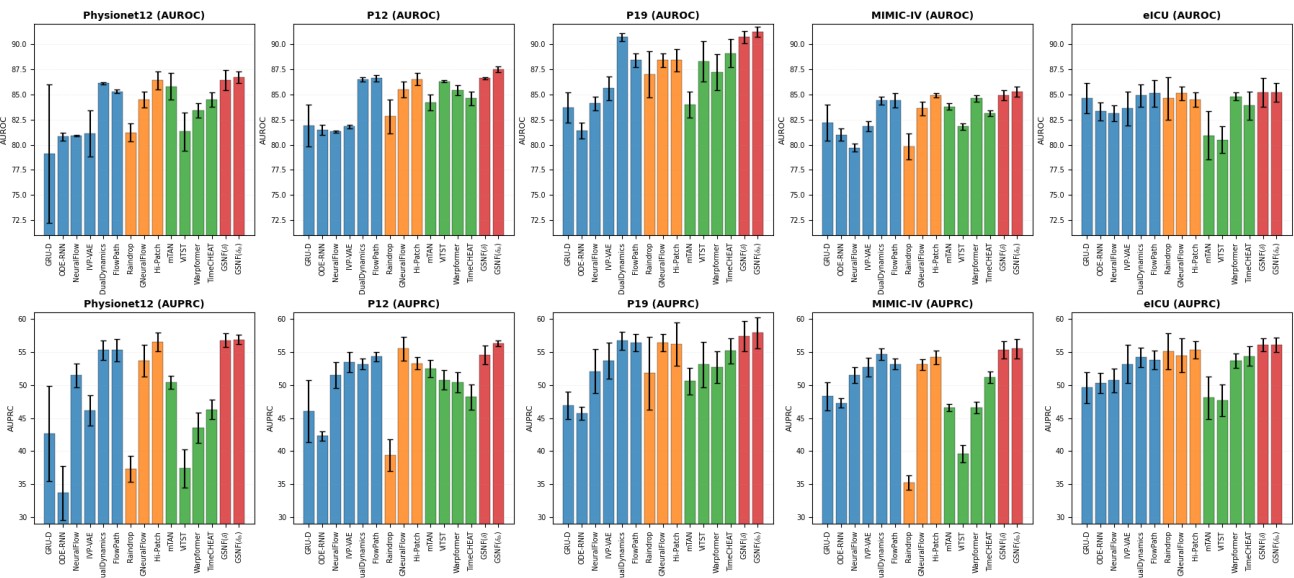

*Figure 8.* Performance variability across five independent runs. Error bars indicate standard deviation.

## E. Experimental Protocol

We conduct comparative experiments using five-fold cross-validation to evaluate GSNF against several representative methods, with AUROC and AUPRC as the primary metrics,

The training workflow of entire model is summarized in Algorithm 1. For testing, the model bypasses loss calculation and directly outputs the predicted labels via a forward pass. All five datasets are used for classification experiments. Each dataset is randomly split into 80% for training, 10% for validation, and 10% for testing. Following previous works (Rubanova et al., 2019; Shukla & Marlin, 2021; Zhang et al., 2022), we repeat each experiment five times using different random seeds to split the datasets and initialize model parameters. For classification experiments, we focus on predicting in-hospital mortality using the first 48 hours of data. Due to class imbalance in these datasets, we assess classification performance using the area under the ROC curve (AUROC) and the area under the precision-recall curve (AUPRC). All models were tested in the same computing environment. The details are as follows:

• Operating System: Ubuntu 22.04.1 LTS

- CPU: Intel(R) Xeon(R) Gold 6330 CPU @ 2.00GHz
- GPU: NVIDIA GeForce RTX 3090 with 24 GB of memory

### E.1. Hyperparameters

For reproducibility, we summarize the hyperparameter settings shared across all experiments, together with the GSNF-specific configurations, in Table 6.

| Hyperparameter | Value | Scope |
|---|---|---|
| Optimizer | Adam | All |
| Weight decay | $1 \times 10^{-4}$ | All |
| Batch size | 50 | All |
| Learning rate (LR) | $1 \times 10^{-3}$ | All |
| LR scheduler step | 20 | All |
| LR decay factor | 0.5 | All |
| Number of GSNF layers | 2 | GSNF |
| Latent dimension | Number of sensors | GSNF |
| Hidden layers | 3 | GSNF |
| Hidden dimension | 128 | GSNF |
| Cross-entropy weight $\alpha$ | 1000 | GSNF |
| ITG weight $\beta$ | 0.1 | GSNF |
| RTG weight $\gamma$ | 0.1 | GSNF |

*Table 6.* Hyperparameter settings for all experiments and GSNF.

### E.2. Datasets and Preprocessing

Our model was evaluated on five representative medical datasets featuring irregularly sampled time series. For consistency, the preprocessing procedures were adopted following the respective referenced works. Detailed information on each dataset and its preprocessing steps is provided below, and the key statistics of the processed datasets are summarized in Table 7.

| | PhysioNet12 | P12 | P19 | MIMIC-IV | eICU |
|---|---|---|---|---|---|
| #Samples | 3,989 | 11,988 | 38,803 | 26,070 | 12,312 |
| #Variables | 37 | 36 | 39 | 96 | 14 |
| Missing ratio (%) | 84.34 | 88.4 | 94.9 | 97.95 | 65.25 |
| Positive rate (%) | 13.89 | 7 | 4 | 13.39 | 17.61 |

*Table 7.* Key information of the five datasets.

The **PhysioNet 2012** dataset (Silva et al., 2012)was released for the PhysioNet/Computing in Cardiology Challenge 2012, aiming to predict in-hospital mortality which is a reduced version of P12 considered by prior work. It contains patient information from ICU admissions, including vital signs, lab tests, and demographic data. In our experiments, we follow Neural Flow (Biloš et al., 2021) and utilize the 4,000 admissions from the challenge's training set, focusing on 37 features recorded within the first 48 hours of each patient's stay.

The **P12** dataset (Goldberger et al., 2000) comprises data from 11,988 patients, including 36 sensor variables and a binary label indicating survival during hospitalization. We used the processed data provided by Raindrop (Zhang et al., 2022).

The **P19** dataset (Reyna et al., 2020) was released for the PhysioNet/Computing in Cardiology Challenge 2019, aiming to predict the onset of sepsis. It contains patient information from ICU stays, comprising static demographics and sparse time-dependent physiological measurements. In our experiments, we utilize 38803 variable-length time series, focusing on 39 features (5 static and 34 time-dependent) recorded within the first 72 hours of each patient's stay for the binary classification of sepsis development.

The **MIMIC-IV** dataset (Johnson et al., 2020) is a multivariate time series dataset composed of sparse and irregularly sampled physiological data collected at the Beth Israel Deaconess Medical Center between 2008 and 2019. Following

a preprocessing approach similar to that of Neural Flow (Biloš et al., 2021), we extract 96 features — including patient intake/output, lab results, and medication prescriptions — from the first 48 hours post-ICU admission. A total of 26,070 patient stays are retained for use in classification tasks.

The **eICU** Collaborative Research Database (Pollard et al., 2018) contains data from patients admitted to ICUs across 208 hospitals in the United States between 2014 and 2015. Following the preprocessing steps outlined by IVP-VAE (Xiao et al., 2024), we extract 14 features within the initial 48 hours post-ICU admission from a total of 12,312 patient stays.

### E.3. Baselines

We compare our model against several baselines for the classification of multivariate irregular time-series.

- **Continuous-time model**:
    - **GRU-D** (Che et al., 2018) incorporates missing patterns using GRU combined with a learnable decay mechanism on both the input sequence and hidden states.
    - **ODE-RNN** (Rubanova et al., 2019) uses an ODE-RNN encoder and Neural ODE decoder in a VAE architecture.
    - **NeuralFlow** (Biloš et al., 2021) model the solution curves directly, with a neural network, instead of specifying the derivative.
    - **IVP-VAE** (Xiao et al., 2024) models irregular time series using a single invertible IVP-based continuous process, eliminating recurrent components and enabling parallel state evolution.
    - **DualDynamics** (Oh et al., 2025b) combines NDE-based method and Neural Flowbased method enhances expressive power.
    - **FlowPath** (Oh et al., 2026) employs an invertible neural flow to learn the geometry of the control path, leveraging invertibility constraints to construct a continuous and data-adaptive manifold for robust modeling of sparse and irregularly-sampled time series.

- **Graph-based models**:
    - **Raindrop** (Zhang et al., 2022) represents dependencies among multivariates with a graph whose connectivity is learned from time series.
    - **GNeuralFlow** (Mercatali et al., 2024) using a directed acyclic graph to model the conditional dependencies of the system components and learning this graph in tandem with neural flow.
    - **Hi-Patch** (Luo et al., 2025) integrates intra-patch graphs for densely sampled local modeling and inter-patch graphs for global multi-scale analysis, leveraging a hierarchical architecture to handle variables with distinct origin scales in Irregular Multivariate Time Series.

- **Other strong baselines**:
    - **mTAN** (Shukla & Marlin, 2021) leverages an attention mechanism to learn temporal similarity and time embeddings.
    - **ViTST** (Li et al., 2023) transforms irregularly sampled time series into line graph images and applies pre-trained vision transformers for classification.
    - **Warpformer** (Zhang et al., 2023) addresses intra-series irregularity and inter-series discrepancy in irregular time series by introducing a warping-based architecture with specialized input encoding.
    - **TimeCHEAT** (Liu et al., 2025) combine channel-dependent modeling at the local (sub-series) level and channel-independent attention at the global level, leveraging bipartite graph-based embedding and Transformer architecture.

