# OpenReview forum: "One-Step Graph-Structured Neural Flows for  Irregular Multivariate Time Series Classification"
_ICML.cc/2026/Conference — ICML 2026 regular_

### Official Review · Reviewer_snq2 · 2026-03-08

**Soundness:** 3
**Presentation:** 3
**Significance:** 3
**Originality:** 3
**Overall Recommendation:** 5
**Confidence:** 4

**Summary:**

This paper proposes Graph-Structured Neural Flows (GSNF), a one-step continuous-time model for irregular multivariate time series that explicitly integrates an interaction graph into the neural flow dynamics. To address the limitation that one-step flows lack iterative refinement, the authors introduce two trajectory-level self-supervision mechanisms: Interaction-Aware Trajectory Generation (ITG), which perturbs the trajectory via intermediate-time re-initialization and enforces a minimum divergence; and Reverse-time Trajectory Generation (RTG), which leverages invertibility to regularize forward-backward consistency.

**Compliance With Llm Reviewing Policy:**

Affirmed.

**Key Questions For Authors:**

1.What is the specific implementation of the "-w/o graph" variant in the ablation study?
2.Does calculating the theoretical $$\delta_{lb}$$ introduce  additional computational overhead?

**Limitations:**

yes

**Strengths And Weaknesses:**

Strengths
1.Novel integration of Neural Flows and graph learning for efficient one-step modeling of complex inter-variable dynamics.
2.Rigorous and interesting mathematical derivation of the data-dependent lower bound $δ_{lb}$ in Eq.  10   for interaction-induced trajectory divergence,  with solid theoretical guarantee and improved performance over manual hyperparameter tuning.
3.Effective self-supervised auxiliary tasks (ITG and RTG) that stabilize and enhance latent interaction graph inference.
4.Robust and SOTA classification performance on diverse time-series benchmarks.
5.Comprehensive and carefully designed ablation experiments validating the effectiveness of ITG, RTG, and core graph modules.
6.Parameter sensitivity analyses with qualitative graph visualizations that improve model interpretability.
7.Clearly methodological presentation, with a reproducibility checklist and public dataset usage.

Weakness
1.The paper provides little discussion on cases where GSNF might underperform. For example, what happens if the interactions are weak or the variables are mostly independent? Does ITG still help, or might it impose spurious distances?
2.In Table 2, the "-w/o graph" variant exhibits a massive performance drop, which is used to justify the necessity of explicit graph modeling. However, it is not entirely clear what architecture replaces the GCN; does it completely abandon the GCN, reducing the neural network in Equation (3) to an MLP? Or does it fix the adjacency matrix to the identity matrix?
3.The sensitivity analysis in Figure 6 is exclusively conducted on PhysioNet12. Although the paper claims that $$\delta$$ is robust and $$\delta_{lb}$$ eliminates the need for tuning, it remains unclear how generalizable $$\delta$$ is, or whether it was tuned separately for each dataset. Table 3 implies that specific datasets have different scales for $$\delta$$ and $$\delta_{lb}$$, but this is not discussed in detail.

---

> ### Author Rebuttal · Authors · 2026-03-31
>
> We sincerely thank Reviewer snq2's valuable feedback.
> 1. **Weak Interaction/Independent  Scenarios (W1)**
>
> Thanks for this interesting question; we will add a Remark in the Appendix.
> - *Weak interaction*: ITG remains valid. ITG lower bound becomes small($\sigma_{min}(A) \to 0$,$\eta = \sigma_{min}(A) \sigma_{min}(W) \Delta_{in}$), but it can still be learned and provides a valid inductive signal.
> - *Independent/No interaction:* if known a priori, A can be set to I, reducing GSNF to a plain neural flow without generating spurious distances. If independence is unknown, ITG may still attempt to learn interactions, potentially producing spurious repulsive distances.
>
> &nbsp;
>
> 2. **Details of the “-w/o graph” variant (W2, Q1)**
>
> In this variant, the adjacency matrix is set to the identity matrix. The GCN structure is retained, but information passing between variables is effectively disabled, so each variable evolves independently along the time axis.
>
> We will explicitly add this detail in Sec. 6.3 to remove any ambiguity.
>
> &nbsp;
>
> 3. **Generalization ability of $\delta$ (W3)**
>
> We thank the reviewer for raising this point. To demonstrate generalizability, we report parameter-tuning results of $\delta$  across all datasets.
>
>
> |  | $\mathsf{Physionet12}$ | | $\mathsf{eICU}$ | | $\mathsf{P12}$ | | $\mathsf{MIMIC\text{-}IV}$ | | $\mathsf{P19}$ | |
> | :--- | :--- | :--- | :--- | :--- | :--- | :--- | :--- | :--- | :--- | :--- |
> | $\delta$ | AUROC | AUPRC | AUROC | AUPRC | AUROC | AUPRC | AUROC | AUPRC | AUROC | AUPRC |
> | 1e-3 | 82.46 | 48.56 | 83.75 | 55.02 | 85.02 | 53.67 | 83.54 | 52.01 | 89.74 | 55.91 |
> | 1e-4 | 82.88 | 48.64 | 83.87 | 54.92 | 85.75 | 54.01 | 83.24 | 52.86 | 90.13 | 56.45 |
> | 1e-5 | 84.89 | 50.38 | **84.05** | **55.48** | **86.24** | **54.55** | 83.56 | 52.61 | **90.76** | **57.32** |
> | 1e-6 | **86.78** | **56.84** | 83.14 | 54.37 | 84.64 | 52.98 | 84.01 | 53.34 | 90.38 | 56.83 |
> | 1e-7 | 84.71 | 49.84 | 82.870 | 53.15 | 83.210 | 51.77 | **84.54** | **54.87** | 89.97 | 56.47 |
> | 1e-8 | 84.35 | 45.73 | 82.550 | 52.86 | 82.540 | 51.54 | 83.78 | 54.63 | 89.51 | 55.92 |
>
>
> Results show that the optimal scale of $\delta$ is dataset-dependent, reflecting differences in interaction strength across datasets. Performance varies significantly with $\delta$, indicating that manual tuning is generally required.
>
> In contrast, our theoretically derived lower bound $\delta_{lb}$ requires no tuning. As shown in Appendix Table 3, the theoretically computed lower bound automatically aligns with the optimal parameter scale for each dataset, and Table 1 demonstrates that its performance actually exceeds that of the manually tuned alternative.
>
> &nbsp;
>
> 4. **Computational overhead of $\delta_{lb}$ (Q2)**
>
> Calculating $\delta_{lb}$ does not introduce additional computational overhead. Its main overhead is the singular value decomposition when calculating $\eta = \sigma_{min}(A)\sigma_{min}(W)\Delta_{in}$（Eq10）, the number of variables in our datasets is extremely small (only 14 to 96)(Tab.6), so the complexity of this operation is effectively at a constant level. In addition, the SVD of the intermediate matrices is just a numerical computation during the training phase and does not participate in backpropagation.

---

> > ### Author Rebuttal · Reviewer_snq2 · 2026-04-01
> >
> > The author solved my problem perfectly, well done! So I raised the rating to 5

---

### Official Review · Reviewer_cYrr · 2026-03-09

**Soundness:** 2
**Presentation:** 3
**Significance:** 3
**Originality:** 2
**Overall Recommendation:** 4
**Confidence:** 3

**Summary:**

In this work, the authors propose an interaction-aware continuous-time modeling framework for irregular multivariate time series, built around efficient one-step flow updates augmented with auxiliary trajectory-level self-supervision (ITG/RTG). The design is intuitive and practical: it aims to retain the computational benefits of one-step flows while injecting graph interactions through additional training objectives. Experiments across five clinical benchmarks show consistent improvements (most notably in AUPRC) while keeping runtime and compute competitive with relevant baselines.

**Compliance With Llm Reviewing Policy:**

Affirmed.

**Final Justification:**

Most of my concerns have been addressed by the authors during the rebuttal, thus I raise my score to Week Accept.

**Key Questions For Authors:**

1.	In Eq. (5), does the model assign a separate adjacency matrix to each segment during training? If yes, why the inference uses a different treatment?
2.	The inferred adjacency matrix has shape $\mathbb{R}^{D_x\times D_x}$, whereas the latent state $\mathbf{z}$ lies in $\mathbb{R}^{D_z}$, how does the method deal with this dimensional mismatch?

**Limitations:**

Pls refer to Weaknesses

**Strengths And Weaknesses:**

**Strength:**

1. The proposed trajectory-level self-supervision tailored for one-step flows: ITG (margin-based divergence via re-initialization) and RTG (consistency via reverse-time decoding), is an interesting and thoughtful response to the absence of solver-based refinement steps.
2. Provides sufficient conditions for invertibility of the proposed flow and a data-dependent lower bound to set the ITG margin without manual tuning.

**Weakness:**

1. The biggest concern is the paper’s static-graph assumption. Classic Graph ODE formulations struggle with time-varying graph structure and therefore typically assume a fixed (static) graph, but this limitation has been addressed by prior work (e.g., [1]). In contrast, this paper already infers segment-specific (dynamic) graphs, yet then collapses them into a single static graph for modeling, which seems difficult to justify and potentially undermines the benefits of learning dynamic connectivity in the first place.
2. The paper’s theoretical framing of the model as a valid ODE solution flow is incomplete: the semigroup property is not established or discussed, and the ITG objective appears to encourage behavior that can conflict with exact ODE flow consistency.
3. The claimed lower-bound derivation for the ITG margin seems weak (or potentially trivial under common normalizations), and the paper does not convincingly argue why this bound is meaningfully non-trivial or practically informative.
4. The related-work discussion does not adequately credit or distinguish prior graph-conditioned neural flow approaches (e.g., GNeuralFlow[2]), which already incorporate graph structure directly into flow dynamics. I think this manuscript risks overstating novelty and should be reframed with clearer empirical/theoretical differentiation.
5. The variational inference of graph structure using a gaussian distribution looks so questionable.

**Reference:**

[1] Tiexin Qin, Benjamin Walker, Terry Lyons, Hong Yan, Haoliang Li. Learning dynamic graph embeddings with neural controlled differential equations. TPAMI, 2025.

[2] Giangiacomo Mercatali, Andre Freitas. Graph Neural Flows for Unveiling Systemic Interactions Among Irregularly Sampled Time Series. NeurIPS, 2024.

---

> ### Author Rebuttal · Authors · 2026-03-31
>
> We sincerely thank Reviewer cYrr's constructive feedback.
> 1. Graph Modeling&Inference_W1,W5,Q1
>
> We would like to clarify the design under latent ODE framework[3].Our model does not aim to recover a discrete ground-truth graph;instead,the “single static graph” is the *initial graph*,serving as the *initial condition of IVP* in the generative model,and thus only one such graph is required.This is not a collapse of dynamic graphs into a static one.
>
> Inferring initial graph is challenging due to *unknown graph structure & highly irregular/missing data*.Prior works using dynamic graphs in ODEs[1] rely on given graph structures.[1] provides dynamic graphs for training & testing.[2] supervises structure only in training.In contrast,we assume unknown graphs in both training & testing under highly irregular observations.
>
> With up to 97% missing, single time points often lack sufficient nodes.Thus,we use segment-level observation to construct "segment-specific graphs".All segments correspond to the same initial graph,and are used only for posterior estimation,not for generative or prediction.
>
> We then infer a posterior over the initial graph and sample a single graph for generation.As no prior graph structure is available,we adopt a Gaussian posterior as a standard and tractable choice in VAE-based inference.The sampled variables are mapped via Softmax to a valid adjacency matrix in (0,1) (Line183).
>
> We acknowledge the value of dynamic graphs; current latent ODEs cannot handle them, but future work may explore dynamic snapshots or other approaches.
>
> ---
> 2. Semigroup property_W2
>
> Semigroup property is *not required* in neural flow ([4,Appendix A.6]) and applies as an additional condition to *autonomous ODEs*. Our flow depends explicitly on t and t_0(Eq2&3),and is *non-autonomous*. We will clarify in revision at Line176.
>
> ---
> 3. ITG is novel & useful_W2,W3
> - Novelty at generation stage. Incorporating interaction structure into generation stage of neural ODEs is highly challenging,as no direct supervision is available[1]. ITG addresses this by introducing a self-supervised signal that captures sensitivity to interaction-induced perturbations, being the first approach in neural flow.
> - Non-triviality of lower bound. The ITG lower bound $\delta_{lb}$ (Theorem 5.1) may appear small in magnitude due to its role as an auxiliary regularization term. However, small magnitude does not imply triviality.Its effectiveness is evidenced by improved performance (Tab1) and more interaction edges (Fig4&5), as also supported by ablation results (Tab2).
> - Practical usefulness(*tuning-free and adaptive*). The bound $\delta_{lb}$ is tuning-free, avoiding costly manual search of $\delta$. In Fig6, $\delta$ is highly sensitive to datasets, while our bound $\delta_{lb}$ adapts automatically during training. Moreover, it is updated jointly with model optimization, providing a finer and more stable signal (Fig7). Empirically, $\delta_{lb}$ matches the scale of the optimal manually tuned $\delta$ (Tab3),while achieving better performance (Tab1).
> ---
> 4. Difference with GNeuralFlow[5]_W4
>
> GNeuralFlow[5] is latent graph-conditioned ODE (inference stage)+plain flow (generation);GSNF(ours) is an *end-to-end graph flow in inference+generation.*
> - Inference stage: GNeuralFlow does not use a flow architecture and relies on an ODE-RNN paradigm.It uses a GNN for initial feature embedding([5,Sec4.2,Eq6),but graph interactions are not propagated into the subsequent dynamics.GSNF adopts a parallel one-step mapping architecture and embeds graph-structured interactions directly into the flow dynamics,allowing interactions to modulate state evolution throughout the entire trajectory.Moreover,while GNeuralFlow requires multiple pretraining epochs to stabilize the DAG-based graph structure,GSNF achieves full end-to-end joint training of both the graph structure and the flow dynamics.
> - Generation stage:GNeuralFlow does not incorporate graph structure in the flow.GSNF incorporates interaction structure into the generation stage via ITG(see discussion in “3.ITG is novel and useful (W2,W3)”),which is highly challenging due to the lack of direct supervision,and enables self-supervised sensitivity to interaction effects that GNeuralFlow cannot provide.
> - Performance:GSNF trains faster(Fig3) and achieves better results(Tab1).
>
> We will add this discussion in Sec5,Line200 to clearly highlight our end-to-end graph flow design and its novelty.
>
> ---
> 5. Dimension_Q2.There is an encoder that outputs $z\in R^{x \times d}$, and will be added at Line200.
> ---
> [1]Learning dynamic graph embeddings with neural controlled differential equations,TPAMI25
>
> [2]Rethink GraphODE Generalization within Coupled Dynamical System,ICML25
>
> [3]Latent ordinary differential equations for irregularly-sampled time series,NeurIPS19
>
> [4]Neural Flows: Efficient Alternative to Neural ODEs,NeurIPS21
>
> [5]Graph Neural Flows for Unveiling Systemic Interactions Among Irregularly Sampled Time Series,NeurIPS24

---

> > ### Author Rebuttal · Reviewer_cYrr · 2026-04-03
> >
> > Thank you to the authors for their efforts during the rebuttal stage; I am still confused about inferring a single graph from the initial state. Based on your claim, the inferred graph would work for the whole segment; thus, this is still a static setup, with all subsequent frames in the segment sharing the same graph structure.

---

> > > ### Author Response · Authors · 2026-04-03
> > >
> > > We sincerely thank Reviewer cYrr for the follow-up question and the opportunity to further clarify the use of a single graph.
> > >
> > > **1. Single Graph as Part of Initial Condition**
> > >
> > > We thank Reviewer cYrr for the careful reading of our rebuttal and for recognizing the role of a single inferred graph in our model.
> > >
> > > We would like to clarify a subtle but important point: the graph $A$ is *not inferred from* initial state $z_0$. Instead,  $(z_0, A)$ jointly form the initial condition. $A$ plays a role analogous to $z_0$.
> > >
> > > &nbsp;
> > >
> > > **2. IVP and Static Setup**
> > >
> > > Reviewer cYrr is correct that this is a static setup, where the inferred graph would work for the whole segment. We emphasize that this is not an additional assumption, but follows naturally from the initial value problem (IVP).
> > >
> > > Let us first recall the classical IVP:
> > >
> > > $\frac{dz(t)}{dt} = f(z(t), t), \quad z(t_0) = z_0,$
> > >
> > > where Picard–Lindelöf theorem [1] guarantees that if $f$ is continuous in $t$ and Lipschitz continuous in $z$, there exists a unique local solution $z(t)$. That is, **the trajectory is uniquely determined by the initial condition $z_0$**.
> > >
> > > Building on the IVP foundation, Neural ODEs[2] compute trajectories sequentially via a solver, while Neural Flows[4] map the initial state directly to the full trajectory in one step. In GSNF, we *extend this to incorporate $A$*:
> > >
> > > $z(t) = F(z_0, A; t), \quad z(t_0) = z_0.$
> > >
> > > | | $z_1$ | $z_2$ | $z_3$ | |
> > > | :--- | :--- | :--- | :--- | :--- |
> > > | Neural ODE[2] | $z_1 = f(z_0; t_1)$ | $z_2 = f(z_1; t_2)$ | $z_3 = f(z_2; t_3)$ | no graph |
> > > | Graph ODE[3] | $z_1 = f(z_0; t_1, A_1)$ | $z_2 = f(z_1; t_2, A_2)$ | $z_3 = f(z_2; t_3, A_3)$ | given graph at each step(may be static or dynamic; see previous rebuttal point 1)|
> > > | Neural Flow[4] | $z_1 = F(z_0; t_1)$ | $z_2 = F(z_0; t_2)$ | $z_3 = F(z_0; t_3)$ | no graph |
> > > | GSNF (ours) | $z_1 = F(z_0, A; t_1)$ | $z_2 = F(z_0, A; t_2)$ | $z_3 = F(z_0, A; t_3)$ | unknown but inferred initial graph; no graph at subsequent steps
> > >  |
> > >
> > > **Key point**: GSNF treats the graph as part of the initial condition, analogous to $z_0$. Unlike Graph ODEs, which require a known graph at each time step, GSNF uses only the initial graph $A$; once $(z_0, A)$ are given, the trajectory is fully determined, with no need for future-step graphs.
> > >
> > > &nbsp;
> > >
> > > **3. Revision Clarification**
> > >
> > > To clarify in our revision, in the current formulation, the graph $A$ is treated as part of the initial condition, analogous to $z_0$. We will update the formulas to make this explicit:
> > >
> > > - **Neural Flow (original):**
> > >   $z(t) = F(z_0, t_0, t)$ (Eq1)
> > >   → **revised:**
> > >   $z(t) = F(z_0, t_0; t)$
> > >
> > > - **GSNF (original):**
> > >   $z(t) = F(z(t_0), t_0, t, A)$ (Eq2)
> > >   → **revised:**
> > >   $z(t) = F(z(t_0), t_0, A; t)$
> > >
> > > This clarifies that both $z_0$ and $A$ are initial conditions, distinct from the target time $t$.
> > >
> > > &nbsp;
> > >
> > > **4. Future work**
> > >
> > > Overall, we thank Reviewer cYrr for raising this point. Extending our model to a time-varying graph $A(t)$ is not a simple extension but a fundamentally different problem. In this setting, the vector field becomes time-dependent, $F(z_0; t, A(t))$, where $A(t)$ is not provided as a known snapshot at each step but must be generated or evolved jointly within $F$. Both node states and graph structure must evolve jointly. One possible direction is to use a coupled system of two flows, with one evolving the node states and the other evolving the graph topology. We leave this as a direction for future work.
> > >
> > > &nbsp;
> > >
> > > **5. Our Contributions**
> > >
> > > Finally, we would like to take this opportunity to briefly reiterate the main contributions of our work.
> > >
> > > To the best of our knowledge, GSNF is the first work to integrate graph-structured interactions directly into end-to-end neural flow framework. Our main contributions are:
> > >
> > > - *Architectural Innovation*. Unlike traditional Graph Neural ODEs which compute sequentially with known graphs, GSNF uses an initial graph and computes all future states in parallel via a neural flow  (Fig1).
> > >
> > > - *Self-supervision Generation via ITG*. As discussed in point 3 of our previous rebuttal, we propose the ITG module, the first approach to incorporate such supervision in neural flow generation, supported by theoretical guarantees and empirically improvements.
> > >
> > > - *SOTA Performance with High Efficiency*. GSNF achieves highly competitive classification performance(Tab1), notably outperforming baselines in AUPRC under extreme missingness, while maintaining efficient (Fig3).
> > >
> > > &nbsp;
> > >
> > > We thank Reviewer cYrr again for the careful and valuable feedback, and would be happy to provide additional clarification if needed.
> > >
> > > &nbsp;
> > >
> > >
> > >
> > > [1] Theory of ordinary differential equations. Tata McGrawHill Education, 1955
> > >
> > > [2] Neural Ordinary Differential Equations, NeurIPS18
> > >
> > > [3] Graph Neural Ordinary Differential Equations, AAAI19
> > >
> > > [4] Neural Flows: Efficient Alternative to Neural ODEs, NeurIPS21

---

### Official Review · Reviewer_1rAC · 2026-03-10

**Soundness:** 3
**Presentation:** 3
**Significance:** 3
**Originality:** 3
**Overall Recommendation:** 4
**Confidence:** 1

**Summary:**

This paper proposes a novel model called GSNF, aiming to address the issue of insufficient modeling of variable interactions in irregular multivariate time series classification. The paper attempts to explore how to more effectively embed and strengthen the graph-structured variable interactions within the one-step mapping architecture of Neural Flows.

**Compliance With Llm Reviewing Policy:**

Affirmed.

**Final Justification:**

My concerns have been addressed, so I keep my positive score.

**Key Questions For Authors:**

None

**Strengths And Weaknesses:**

Strengths:
A framework has been proposed that directly integrates graph interaction into a one-step mapping flow model, breaking the previous limitation where models only utilized graph structures as initial conditions. It achieved state-of-the-art classification performance on five real-world datasets, and was highly competitive in terms of training time and memory usage.

Weaknesses:
Although various SOTA models were compared, due to its architecture being based on ResNet flow, further discussion can be conducted regarding the upper limit of the expression capability of this one-step mapping architecture when dealing with extremely long sequences or systems with higher dynamical complexity.

---

> ### Author Rebuttal · Authors · 2026-03-31
>
> Thank you for Reviewer 1rAC's insightful comments.
>
> Regarding the expression capability of the one-step mapping architecture, as noted in [1, Sec 2.2], "*The ResNet flow...can be viewed as an Euler discretization, meaning it is enough to stack appropriately many layers to uniformly approximate any ODE solution.*" This indicates that the one-step formulation does not inherently limit model expressiveness, as capacity is governed by network depth.
>
> However, due to the ODE-based nature of the model, we agree that handling extremely long sequences or highly complex dynamics remains challenging, especially compared to context- and attention-based Transformer methods or state-space models. At the same time, ODE-based models are better suited for continuous-time modeling, especially under irregular sampling and non-ideal observations.
>
> This limitation is not specific to the one-step formulation, but is common to ODE-based continuous-time models. We will clarify this point and include it in the limitations (Sec. 8, Impact Statements) of the revised paper.
>
> &nbsp;
>
> [1] Neural Flows: Efficient Alternative to Neural ODEs, NeurIPS'21

---

> > ### Author Rebuttal · Reviewer_1rAC · 2026-04-01
> >
> > I have maintained my score.

---

### Decision · Program_Chairs · 2026-04-30

**Decision:**

Accept (regular)

**Comment:**

This submission proposes Graph-Structured Neural Flows (GSNF) for irregular multivariate time-series classification. The core idea is to incorporate a learned interaction graph directly into a one-step neural flow, and to strengthen interaction learning with two auxiliary trajectory-level objectives, interaction-aware trajectory generation (ITG) and reverse-time trajectory generation (RTG). The empirical evaluation is broad, covering five clinical benchmarks, and the paper reports that GSNF variants achieve the top results across AUROC/AUPRC while remaining computationally competitive; in particular, the theoretically derived margin variant outperforms the manually tuned variant on four of five datasets and does so without meaningful extra overhead.

The reviewer discussion was overall positive. Reviewers, in general, appreciate the technical novelty while raising some concerns, which had been mostly addressed during the rebuttal period. Overall, I find the paper to make a meaningful contribution at the intersection of efficient continuous-time modeling and graph-structured irregular time-series learning. For the final version, I strongly encourage the authors to make the static-graph scope explicit, narrow the theoretical wording so that claims align with the proved properties, and strengthen the comparison to GNeuralFlow and related graph-conditioned continuous-time approaches.